# Influence of Restrained Condition on Mechanical Properties, Frost Resistance, and Carbonation Resistance of Expansive Concrete

**DOI:** 10.3390/ma13092136

**Published:** 2020-05-05

**Authors:** Nguyen Duc Van, Emika Kuroiwa, Jihoon Kim, Hyeonggil Choi, Yukio Hama

**Affiliations:** 1Division of Engineering, Muroran Institute of Technology, Muroran, Hokkaido 050-8585, Japan; 18096012@mmm.muroran-it.ac.jp; 2Course of Architecture and Building Engineering, Muroran Institute of Technology, Muroran, Hokkaido 050-8585, Japan; 20041026@mmm.muroran-it.ac.jp; 3College of Environmental Technology, Muroran Institute of Technology, Muroran, Hokkaido 050-8585, Japan; bmjhun@mmm.muroran-it.ac.jp; 4School of Architecture, Kyungpook National University, Daegu 41566, Korea

**Keywords:** expansive concrete, restrained condition, expansion strain, pore structure, frost resistance, carbonation resistance

## Abstract

This paper presents the results of an experimental investigation of the effect of the restrained condition on the mechanical properties, frost resistance, and carbonation resistance of expansive concrete with different water–binder ratios. In this study, length change ratio test, expansion strain test, compressive strength test, mercury intrusion porosimetry test, underwater weighing test, freezing–thawing test, and accelerated carbonation test were performed to evaluate the mechanical properties, pore size distribution, total porosity, and durability of expansive concrete under both restrained and unrestrained conditions. The test results indicate that the length change ratio and expansion strain of the expansive concrete were controlled by the restrained condition. The compressive strength of expansive concrete was enhanced by the triaxial restraining when the amount of expansive additive was 40 kg/m^3^ of concrete. Two hypotheses were described to explain the change of pore structure change expansive mortar. The results also indicate that the carbonation resistance and frost resistance were improved by the uniaxial restrained condition. Furthermore, the effect of the restrained condition must be considered to evaluate not only the experimental results of the expansive concrete with a high EX replacement level but also the expansive concrete combining other cement replacement materials.

## 1. Introduction

Concrete is one of the most common widely used materials in structure construction because of its advantages, such as high performance, durability, and low cost. However, concrete has a major unfavorable property, which is drying shrinkage [1]. It is well known that drying shrinkage is generated by the loss of capillary water from the hardened cement mixture, which causes an increase in tensile stress, leading to the formation of crack. The cracks caused by drying shrinkage leads to reduced strength and durability and increased risk of corrosion. Therefore, it is necessary to protect the concrete structure from the drying shrinkage to improve the durability and the lifetime of the structure. To address this problem, expansive concrete is widely used to efficiently prevent shrinkage cracks. Expansive concrete is nowadays made in two ways. The first method is using expansive cement to make expansive concrete. The second approach is using an expansive additive as a cement replacement material to make expansive concrete. The latter method has been widely used in Japan and Thailand [2]. According to the American Concrete Institute standard [3], three kinds of expansive cement are available: K, M, and S types. According to the JIS A 6202 [4], an expansive additive is an admixture that reacts with water and makes the concrete expand due to the production of ettringite and calcium hydroxide from the hydration reaction.

Expansive concrete has been applied to slabs, pavements, shrinkage-compensating concrete structures, etc. [5]. Considerable laboratory research on the mechanical properties and durability of expansive concrete has been undertaken. However, these experiments were performed under unrestrained (free) conditions (e.g., compressive strength, accelerated carbonation, and freezing–thawing tests). In an actual concrete structure, the expansive concrete is usually restrained by reinforcement, connecting members, foundation, etc. Therefore, the behavior of expansive concrete under restrained conditions must be investigated in a laboratory.

Several studies were conducted to investigate the effect of restrained conditions on the expansive concrete properties. Tsuji et al. [6] reported that the compressive strength of expansive concrete was higher than that of cement concrete by approximately 30%–50% under a restrained condition. The amount of expansive additive (EX) was used at a high level, which was approximately 80 kg/m^3^ of concrete. However, as a standard amount in Japan, EX is known as 20 kg/m^3^ of concrete. Similar to the results reported by Tsuji et al., Nguyen et al. [2] also reported that the compressive strength of expansive concrete could be improved by confinement. However, the expansive concrete confinement was only restrained by the horizontal directions and did not consider the vertical directions. Considering the effect of restrained conditions, Harada [7] found that the frost resistance of the restrained concrete was enhanced compared to that of the unrestrained conditions for cement concrete. Furthermore, Colin et al. [8] found a decrease in the size of the large pores of expansive cement paste under uniaxial restrained conditions. Note that most of these works only focused on the mechanical behavior of expansive concrete under uniaxial restrained conditions. Meanwhile, an experimental study on the effect of both uniaxial and triaxial restrained conditions on the mechanical properties and durability of expansive concrete has not yet been established.

Therefore, the main objectives of this research are: firstly, studying the influence of both uniaxial and triaxial retrained conditions on the mechanical properties, pore structure change, carbonation resistance and frost resistance of expansive concrete; secondly, grasping these effects to obtain a correct assessment of the experiments on expansive concrete in a laboratory. The mold was designed to restraint the expansion of the expansive concrete in a triaxial restraint for the compressive strength test. A restrained mold was used to evaluate the length change ratio and the expansion strain of cement and expansive concrete according to JIS A 6202 [4]. The underwater weighing and mercury intrusion porosimetry (MIP) tests were used to monitor the change of the total porosity and the pore size distribution of the expansive concrete mixtures under the free and restrained conditions. In addition, the effect of the restrained condition on the carbonation and frost resistance of expansive concrete was investigated using a designed mold according to Hanara’s method [7].

## 2. Experimental Program

### 2.1. Experimental Materials and Mix Proportions

The material properties are arranged in Table 1. Ordinary Portland cement, CSA-type expansive additive (calcium sulfoaluminate), JIS standard aggregate (coarse and fine aggregates), and tap water were used. An AE water-reducing agent (Master Pozzolith No. 70), high-performance water-reducing agents, and an AE agent (Master Air 101) were used to control the concrete workability and air content.

Table 2 provides the mix concrete proportions. The amount of EX replacing the binder was 0, 20, and 40 kg/m^3^. Concrete and mortar were prepared with water–binder ratios of 0.3 and 0.5, respectively. The concrete samples were used for the tests on the length change ratio, expansion strain, underwater weighing, compressive strength, and accelerated carbonation under both free and restrained conditions. The mortar samples were used to determine the pore size distribution using the MIP test.

### 2.2. Test Methods

#### 2.2.1. Length Change Ratio and Expansion Strain Tests

The length change ratio and expansion strain of concrete tests were performed under both free and restrained conditions. For the free condition, the free expansion of concrete was conducted in accordance with the JIS A 1129 [9]. For this test, steel prisms measuring 100 × 100 × 400 mm^3^ and 75 × 75 × 400 mm^3^ were used. The specimens were demolded at 24 h after casting to avoid steel mold confinement. An embedded strain gauge was installed into the center of the unstrained specimens to measure the expansion strain for the free condition (Figure 1a and Figure 2a). For the restrained condition, the concrete was cast into the restrained molds according to the JIS A 6202 [4] (Figure 1b). A strain gauge was attached on the top and bottom surface of the rebar at the center of the specimen. A data logger was used to immediately record the strains of the unrestrained and restrained concrete after casting. Additionally, a restrained method was designed with a size of 75 × 75 ×400 mm^3^ according to the free standard (Figure 2b) to evaluate the effect of different restrained conditions on concrete expansion.

#### 2.2.2. Mercury Intrusion Porosimetry Test

The interfacial transition zone is well known to have a significant effect on the pore size distribution of the hardened concrete. The MIP test was conducted on the mortar samples herein to address this problem. The samples were obtained by separating the coarse aggregates from the fresh concrete by wet sieving (5 mm sieve). The unrestrained mortar was cast in prism molds measuring 40 × 40 × 160 mm^3^ and demolded after 1 day. Sealed curing was then performed in a room at a temperature of 20 ℃ and relative humidity of 60%. On the contrary, the restrained mortar was cast into the designed molds (Figure 3). At the test ages, the mortars were cut into small 5× 5× 5 mm^3^ cubes and soaked in ethanol for 1 week prior to being vacuum freeze-dried for 24 h to stop the hydration. The mortar samples for this test were randomly selected.

#### 2.2.3. Underwater Weighing Test

Many studies [2,10,11,12] used the MIP test to determine the total porosity of mortar and concrete. However, the MIP test cannot measure a pore size (diameter) smaller than 3 nm [8,13]. Meanwhile, the gel pore diameter ranges from 1 nm to 10 nm [13]. Therefore, the underwater weighing (Figure 4) test was used herein to determine the satisfactory value of the total porosity of concrete. The preparation of the total-porosity test samples was similar to that for the MIP sample test. After being vacuum freeze-dried for 24 h, the cube samples were immersed in water and continued to be kept in a vacuum chamber for 24 h. The masses of the cube samples were determined under the water in a saturated surface-dry condition and after drying at 105  °C for 24 h. The total porosity was calculated as follows:(1)$$V_{t}\;\left(\%\right)=\left(1-\frac{\rho_{b}}{\rho_{tr}}\right)\times 100\%$$
(2)$$\rho_{b}\;\left(g/{\mathrm{cm}}^{3}\right)=\frac{m_{oven}}{m_{sa}-m_{water}}\times \rho_{w}$$
(3)$$\rho_{tr}\;\left(g/{\mathrm{cm}}^{3}\right)=\frac{m_{oven}}{m_{oven}-m_{water}}\times \rho_{w}$$
where $V_{t}$ is the total porosity (%); $\rho_{b}$ is the bulk density (g/cm^3^); $\rho_{tr}$ is the true density (g/cm^3^); $\rho_{w}$ is the density of water (g/cm^3^); $m_{oven}$ is the sample mass after drying at 105 °C; $m_{sa}$ is the sample mass of saturated surface-dried specimen (g); $m_{water}$ = $m_{t}-m_{ap}$ is the mass of sample underwater (g); $m_{t}$ is the mass of the equipment and sample underwater (g); $m_{ap}$ is the mass of the equipment underwater (g).

#### 2.2.4. Compressive Strength Test

The compressive strength was measured on ∅100 × 200 mm^2^ cylinders at different conditions (i.e., unrestrained and restrained conditions) to evaluate the effect of the restrained condition on the compressive strength of concrete. The unrestrained concrete was demolded 24 h after casting and covered by a plastic sheet to cure at 20 ℃ and 60% relative humidity until the time of the test. For the restrained condition, Nguyen et al. [2] designed a cube mold measuring 100 × 100 × 100 mm^3^, which was restrained using two steel bars to connect with a steel plate at the specimens’ ends. However, these molds could only confine the concrete in one direction (i.e., horizontal direction) and did not consider the vertical direction. To address this limitation, the concrete was cast in the steel molds to restraint three directions. The vertical direction was restrained by connecting the two-steel plate using four steel bars (Figure 3). Both the unrestrained and restrained compressive strengths were tested in accordance with JIS A 1108 [14] at 3, 7, and 28 days.

#### 2.2.5. Freeze–thaw Test

In this test, the frost resistance of concrete was determined according to the JIS A 1148 [15] with dimensions of 75 × 75 × 400 mm^3^ under the free and restrained conditions (Figure 2). The fundamental transverse frequency and the mass loss change of the specimens were measured within 300 cycles. The resistance of concrete to freeze–thawing was evaluated by calculating the value of the relative dynamic modulus of elasticity (RDM), as given in Equation (4). The specimens are often considered as frost damage when the RDM value is less than 60%.
(4)$$P_{n}=\left(\frac{f_{n}^{2}}{f_{0}^{2}}\right)\times 100\%$$
where $P_{n}$ is the relative dynamic modulus of elasticity (%); $f_{n}$ is the fundamental transverse frequency at *n* cycle (Hz); $f_{0}$ is the fundamental transverse frequency at 0 cycles (Hz).

#### 2.2.6. Accelerated Carbonation Test

The accelerated carbonation test was conducted in accordance with JIS A 1153 [16] to measure the carbonation depth of all the concrete mixtures measuring 100 × 100× 400 mm^3^. The carbonation depth was measured by spraying a phenolphthalein solution on the test surface after 13 weeks of exposure to an accelerated carbonation condition (ambient temperature: 20 ± 2 °C; CO_2_ concentration: 5  ± 0.2%; and relative humidity: 60 ± 5%).

## 3. Results and Discussion

### 3.1. Fresh Concrete

Table 3 presents the test results of the fresh non-expansive and expansive concrete. The slump, flow, and air content results were obtained in accordance with the JIS A 1101 [17], JIS A 1150 [18], and JIS A 1128 [19], respectively. Table 3 shows that the slump and the flow of concrete were almost the same for all samples in the two series. The tendency was also seen in the air content results for concrete with w/b 0.5. Meanwhile, the air content of the fresh concrete with w/b 0.3 slightly increased when the amount of expansive additive was increased. These tendencies were similar to those reported by M. Tsuji et al. [20], who showed the results of a new test method for the restrained expansion of expansive concrete.

### 3.2. Length Change Ratio and Expansion

Figure 5 presents the length change ratio of the cement and expansive concrete with different w/b ratios and types of retrained conditions. In the case of w/b 0.3 (Figure 5a–c), an insignificant effect of the restrained condition was observed on the length change ratio of both cement and expansive concrete. Moreover, these results indicate that the length change ratio increased when the expansive additive dosage was increased. These tendencies were found in the test results of w/b 0.5 (Figure 5d–f). However, the effect of the restrained condition on the length change ratio was observed in the case of w/b 0.5, especially for EX40. The length change ratio of the restrained concrete was smaller than that of the unrestrained concrete and was clearly observed when the amount of the expansive additive reached 40 kg/m^3^. When different restrained conditions were considered (free standard, Figure 5d and e; JIS A 6202, Figure 5f), the expansive concrete sample was tested following the JIS A 6202. It showed the smallest length change, which was caused by the restraint of the steel bars inside it and the two steel plates at both its ends (Figure 1b). Meanwhile, the expansive concrete sample was tested according to the free standard, which was restrained by reinforcement (Figure 2b). The expansive concrete with w/b 0.3 had a smaller length change ratio compared to the expansive concrete with w/b 0.5. This result may be explained by the concrete stiffness. The concrete with w/b 0.5 had a smaller stiffness that led to a higher expansion of the concrete containing the expansive additive. This explanation is consistent with the hypothesis reported by Nguyen et al. [2].

Figure 6 shows the expansion strain of the non-expansive and expansive concrete with different w/b ratios and types of restrained conditions. The expansion strain tendency was similar to the length change ratio of concrete (Figure 5). However, unlike in the test results of the length change ratio of concrete with w/b 0.3, the effect of the restrained condition on the expansion strain of the expansive concrete was observed, which may be attributed to the different testing positions. Meanwhile, the length change of the sample was determined by measuring the change of the whole sample. The expansion strain test was performed on the inside of a sample. Relevant to the expansion strain of EX40 with different types of restrained conditions (Figure 6e,f), the expansion strain of concrete restrained by the JIS A 6202 was higher than that restrained by the free standard. The higher expansion strain of EX40 by the JIS method can be explained as follows: the expansion strain of concrete was determined by the strain gauge attached to the steel bar surface at the center of the specimen to measure the steel bar strain. In other words, the steel bar expanded (shrunk) with the expansion (or shrinking) of concrete. Therefore, when the expansive concrete was restrained by the JIS method, the effect of the expansion force on the steel bar was higher than that on the steel bar of free standard.

The results demonstrate that the expansive concrete expansion could be restrained by restraining a subject, such as a rebar. The rebar ratios and the degree of restraint could also affect the expansive concrete expansion. This study focused on a uniaxial restrained condition for the expansion strain test. Hence, studying the influence of the degree of restraint on the expansive concrete expansion will be very interesting in the future.

### 3.3. Pore Structure

Figure 7 shows the test results of the pore size distribution of the cement mortar (OPC sample) and the expansive mortar (EX20 and EX40) with w/b 0.3 and 0.5 obtained by the MIP test at 28 days. The pore size distribution curves of the control mortar sample (OPC sample) typically exhibited at least two sharp peaks for w/b 0.3 and 0.5. The first one laid less than 10 nm, while the second corresponded to a diameter of approximately 20–50 nm. The first peak that laid less than 10 nm was difficult to observe in the expansive mortar. The results showed that the pore diameter corresponding to the highest peak in the differential curve slightly decreased with the addition of the expansive additives for both w/b 0.3 and 0.5. When comparing different w/b ratios, these pore diameters increased with a higher w/b ratio. For the EX40 sample with both w/b 0.3 and 0.5, the rounded peak appeared in the pore size ranging from 100 to 1000 nm, indicating that the EX addition affected the pore diameter of 100–1000 nm. However, the effect of the restrained condition on this pore diameter range was not observed.

The concept of a critical pore entry diameter was considered to evaluate the influence of the restrained condition on the pore size distribution of the control and expansive mortar samples. Scrivener et al. [21] reported that the critical pore entry diameter is the pore diameter, where the steepest slope of the cumulative curve is recorded. The critical pore entry diameter is the pore diameter that corresponds to the highest peak of the differential curve. Table 4 shows the test results of the critical pore entry diameter. Under the same w/b ratio, the critical pore entry diameter increased with the increasing amount of EX. The critical pore entry diameter was decreased by the restrained condition. In other words, under the restrained conditions, the pore diameter tended to become finer, which may be explained by the hypotheses below.

The first hypothesis is relevant to the formation of the ettringite crystal when the expansive additive reacts with water, which will fill the microspore in the cement–expansive paste matrix. This phenomenon leads to a denser pore structure. This hypothesis was also reported by Van et al. [22] and Nguyen et al. [2].

For the second hypothesis, the pore size distribution of the restrained sample shifts to a finer region because of the chemical stress effect [6]. The expansive force is caused by the EX hydration that makes the pore diameter become smaller by the constraining pressure of the external constraining restrained mold (e.g., framework) (Figure 8).

### 3.4. Total Porosity

The under-water weighing test was used herein to determine the total porosity of concrete under the free and triaxial restrained conditions. Figure 9 illustrates the total porosity results for different concrete mixtures at 28 days. Consequently, the total porosity slightly increased as the amount of EX increased. This result is consistent with the findings of Choi et al. [23], who reported that the volume of the capillary pore increases with the increasing EX addition. The total porosity of the concrete with and without EX at the w/b ratio of 0.5 was higher than that at the w/b ratio of 0.3. The total porosity of the restrained sample at the w/b ratio of 0.5 was approximately 3% and 21% less than that of the unrestrained samples for EX20 and EX40, respectively. In other words, the pore structure of the expansive concrete was denser under the restrained condition. However, this tendency was not observed for EX40 at the w/b ratio of 0.3, which can be attributed to the higher concrete stiffness at a lower w/b ratio making the concrete expand more hardly, thereby leading to a decrease in the effect of the restrained condition. In conclusion, observing the effect of the restrained condition on the total porosity was easier with a lower stiffness in the expansive concrete.

### 3.5. Compressive Strength of Concrete

The compressive strengths of cement and expansive concrete were tested under the restrained and unrestrained conditions. Figure 10 illustrates the compressive strength results of all the concrete mixtures at the ages of 3, 7, and 28 days of sealed curing age. The strength values were the average of three test samples.

Figure 10 shows that the compressive strength of concrete varied between 45 MPa to 130 MPa and 20 MPa to 53 MPa for the w/b ratios of 0.3 and 0.5, respectively. The compressive strength of concrete slightly decreased with the increasing amount of EX because the total porosity increased with the increasing EX addition, which decreased the compressive strength. This result is consistent with those of the previous studies [22,23,24], which showed that the compressive strength decreases as the total porosity increases. The compressive strength of concrete at the w/b ratio of 0.3 was not enhanced by the restrained condition. This phenomenon was also found in the results of OPC and EX20 at the w/b ratio of 0.5. However, the effect of the restrained condition on the compressive strength was clearly observed for EX40 at the w/b ratio of 0.5. The results indicate that the compressive strength of the restrained concrete was 12% larger than that of the unrestrained concrete and was approximately similar to that of EX20 at 28 days. The results also imply that at an early age of 3 days, the EX40 restrained concrete showed a larger compressive strength than both EX20 restrained and unrestrained concrete. This result is possibly caused by the decrease in the total porosity of the expansive concrete (Figure 9b) under the confinement condition, which led to its higher strength. These results demonstrate that the compressive strength of the expansive concrete was improved under the restrained conditions as the amount of EX increased to 40 kg/m^3^. The conclusion corresponded to the results of Nagataki et al. [25], who studied the relations among unit expansive additive content, compressive strength, and restrained expansion rate.

### 3.6. Frost Resistance of Concrete

Figure 11 shows a comparison of the RDMs of the Portland cement and expansive concrete under the restrained and unrestrained conditions at the w/b of 0.3 and 0.5. The values of the result were the average of two test samples.

The RDM decreased with the increasing amount of expansive additive in both the restrained and unrestrained conditions. The results demonstrate that the presence of EX led to a reduction of the frost resistance of concrete, especially for the unrestrained concrete. This result is consistent with that of the previous study [26] and may be explained by the formation of the micro-cracks caused by the expansion of the EX hydration under the free conditions. When evaluating the effect of the restrained condition of the frost resistance of concrete, the results indicated that for the w/b ratio of 0.3, the restrained concretes had a higher RDM than the unrestrained concretes because the RDMs of OPC, EX20, and EX40 were approximately 80% and 87% at 300 cycles. For the w/b ratio of 0.5, observing the change of the RDM of the OPC and EX20 was difficult. However, the expansive concrete (EX40) showed early frost damage with an RDM below 80% at 12 freezing–thawing cycles and below 60% at 100 cycles under free condition. Meanwhile, the RDM of EX40 was below 60% at 190 cycles under the restrained condition.

The results conclude that the frost resistance of the expansive concrete was enhanced by the restrained conditions at the lower w/b ratio. Furthermore, a previous study [7] found that under the triaxial restraint, the frost resistance of the Portland cement concrete was better than that of the uniaxial restrained concrete. According to this conclusion, the present research may be extended in the future by investigating the influence of the degree of restraint on the frost resistance of expansive concrete, especially the triaxial restraint.

Figure 12 plots the mass change of all concrete mixtures under the restrained and unrestrained conditions following the test cycle. Similar to the relative dynamic modulus of elasticity, the weight of the specimens at the w/b ratio of 0.5 decreased with the increasing EX dosage at 300 cycles. The effect of the confinement was only observed for EX40 as the weight loss of the unrestrained concrete became slightly larger than that of the restrained concrete. The weight loss percentage was not observed for all the samples at the w/b ratio of 0.3, which may be explained by the dense structure of the concrete at the lower w/b ratio. However, the mass change was observed. Figure 12a shows that from 0 to 200 cycles, the mass change of the samples increased when the EX replacement weight was increased because the capillary pore volume in the expansive concrete was larger than that of the cement concrete [23], thereby leading to the expansive concrete easily absorbing water. However, from 200 to 300 cycles, the results showed an opposite trend as the mass change decreased when the EX amount was increased. This finding can be attributed to the increase in the scaling on the sample surface. The results also showed that the effect of the restrained condition on the weight loss percentage of all samples was hardly observed.

### 3.7. Carbonation Depth of Concrete

The carbonation depth of concrete with and without expansive additive at different w/b ratios was measured through the phenolphthalein test exposed in the controlled condition (temperature: 20 ℃; relative humidity: 60%; and CO_2_ concentration: 5%) for 13 weeks (Table 5). The concrete carbonation depth results were the average of 18 to 20 measurements from the exposed surface. The test results indicated that concrete was not carbonated under a w/b ratio of 0.3 compared with that under a w/b ratio of 0.5. This result may be attributed to the compact structure of the concrete at the lower w/b ratio. The concrete carbonation depth increased with the increasing w/b ratio. This observation is consistent with that reported by Singh et al. [27]. At the w/b ratio of 0.5, the carbonation depth increased with the addition of the expansive additive. However, comparing the different restrained conditions, the carbonation depth of the restrained condition was smaller than that of the free condition. This result can be explained by the denser structure of the restrained concrete caused by the compressed pore in the concrete by the chemical prestress effect. This phenomenon was discussed in Section 3.3.

The results demonstrate that the carbonation resistance of the expansive concrete was enhanced by restraining. The concrete carbonation specimen was made under the uniaxial restrained condition. As mentioned, the chemical prestress effect made the pore structure of the expansive concrete denser under the restrained conditions. Nagataki et al. [25] also reported that the chemical prestress effect greatly depends on many factors, such as kind of cement, w/b ratio, curing method, reinforcing rebar ratio, and degree of restraint. Therefore, in the future, more investigations are necessary to find the influence of these factors on the chemical prestress effect in expansive concrete.

## 4. Conclusions

The following major conclusions are drawn based on the results and discussion in this study:(1)The length change ratio and expansion strain of concrete with and without an expansive additive were controlled by the restrained condition.(2)Under the restrained condition, the pore size distribution of the expansive mortar shifted to a smaller size, which can be explained by two hypotheses. First, the microspore in the cement–expansive paste is filled with crystal ettringite. Second, the effect of chemical stress is caused by the expansive additive hydration.(3)The total porosity of the expansive concrete was decreased by the restraining when the amount of EX was 40 kg/m^3^ of concrete at a w/b ratio of 0.5. The compressive strength of the restrained concrete was 12% higher than that of the unrestrained concrete at 28 days.(4)The enhanced frost resistance of the expansive concrete was clearly observed at the w/b ratio of 0.3. However, at the w/b ratio of 0.5, the frost resistance of the expansive concrete was enhanced by restraining for EX40.(5)The concrete carbonation depth was not observed at the w/b ratio of 0.3. For the w/b of 0.5, the carbonation resistance of the expansive concrete was improved by restraining.(6)It is essential considering the effect of the restrained conditions in evaluating the various experimental results of the expansive concrete with a large expansive additive dosage (i.e., over 20 kg/m^3^ concrete) or concrete combining EX with other cement replacement materials (e.g., fly ash and blast furnace slag).

## Figures and Tables

**Figure 1 materials-13-02136-f001:**
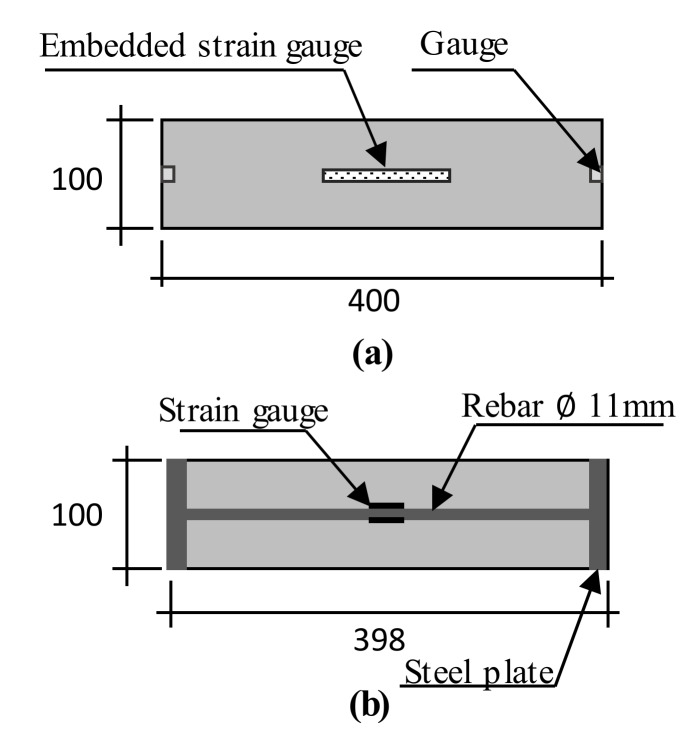
Schematic of the specimens measuring 100 × 100 × 400 mm^3^: (**a**) free condition and (**b**) restrained condition according to the JIS A 6202 [4].

**Figure 2 materials-13-02136-f002:**
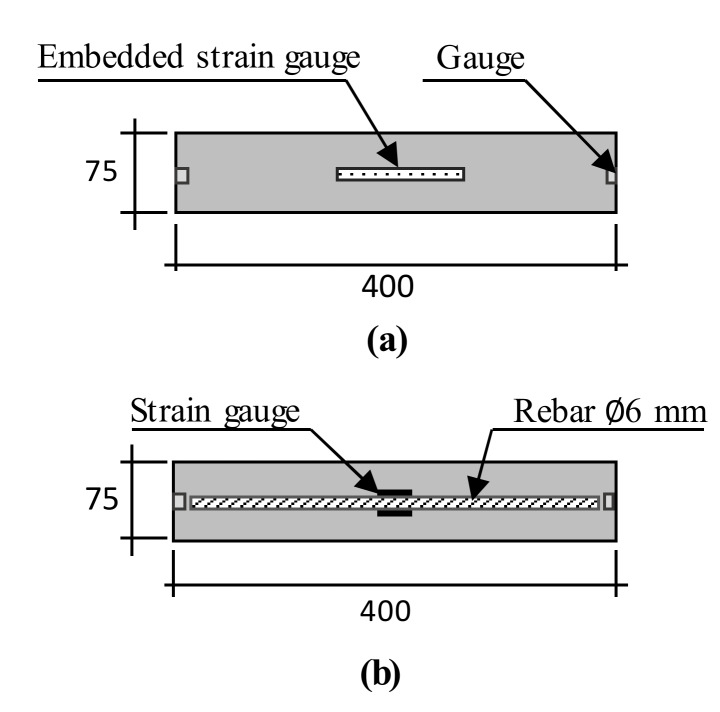
Schematic of the specimens measuring 75 × 75 × 400 mm^3^: (**a**) free condition and (**b**) restrained condition [7].

**Figure 3 materials-13-02136-f003:**
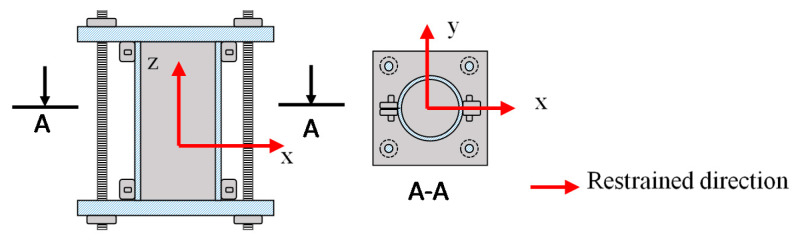
Schematic of the restrained mold for testing the pore structure and restrained compressive strength.

**Figure 4 materials-13-02136-f004:**
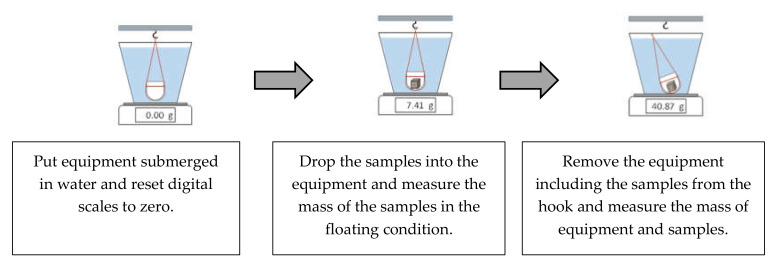
Schematic of the underwater weighing test.

**Figure 5 materials-13-02136-f005:**
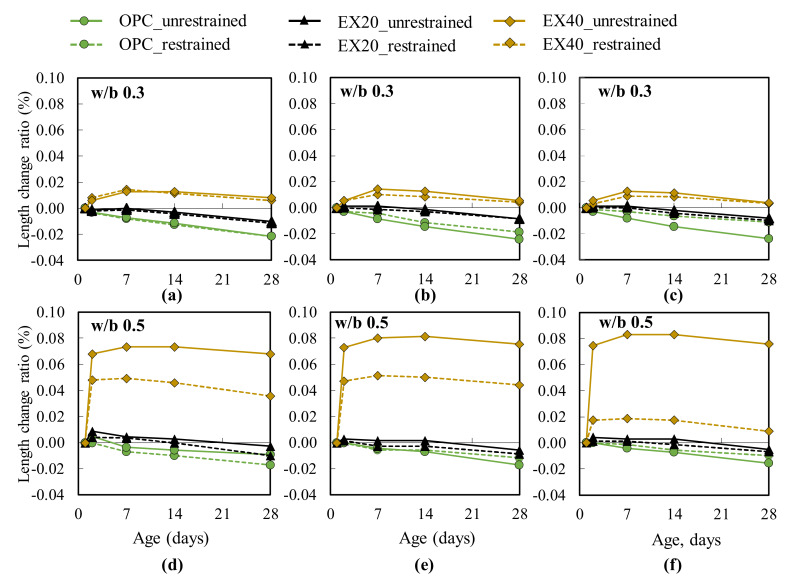
Length change ratio of concrete: (**a**,**d**) results of the sample measuring 75 × 75 × 400 mm^3^; (**b**,**e**) results of the sample measuring 100 × 100 × 400 mm^3^ (free standard); (**c**,**f**) results of the sample measuring 100 × 100 × 400 mm^3^ (JIS A 6202).

**Figure 6 materials-13-02136-f006:**
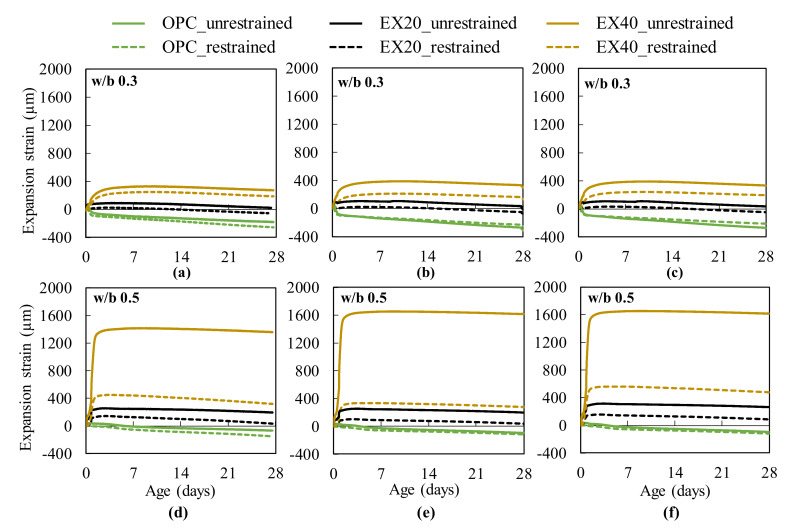
Expansion strain of concrete: (**a**,**d**) test results of the sample measuring 75 × 75 × 400 mm^3^; (**b**,**e**) test results of the sample measuring 100 × 100 × 400 mm^3^ (free standard); and (**c**,**f**) test results of the sample measuring 100 × 100 × 400 mm^3^ (JIS A 6202 [4]).

**Figure 7 materials-13-02136-f007:**
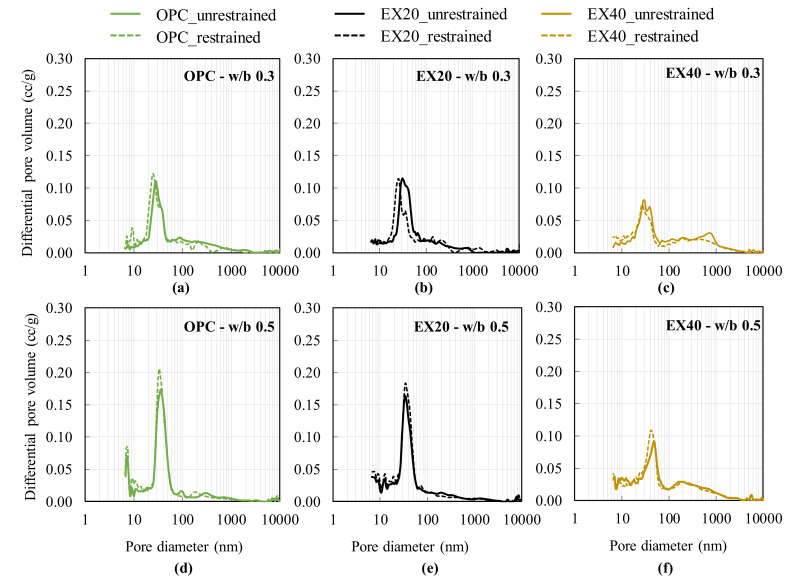
Effect of the triaxial restrained condition on the pore size distribution of the cement and expansive mortar under the restrained and unrestrained conditions with different w/b ratios: (**a**–**c**) test results of OPC, EX20, and EX40 with w/b 0.3, respectively; and (**d**–**f**) test results of OPC, EX20, and EX40 with w/b 0.5, respectively.

**Figure 8 materials-13-02136-f008:**
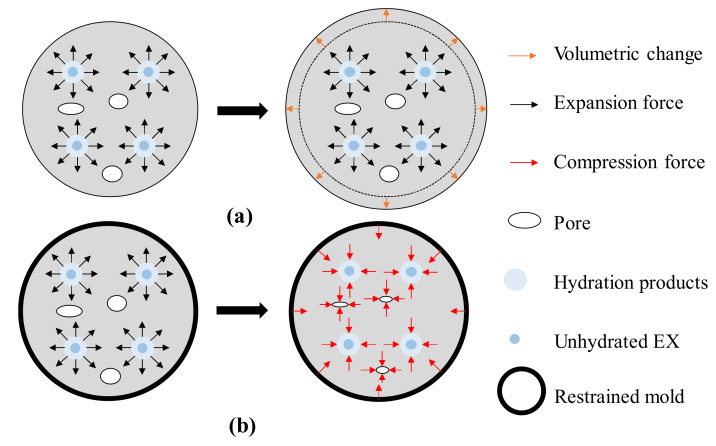
Schematic of the effect of chemical stress on the pore structure: (**a**) free condition and (**b**) triaxial restrained condition.

**Figure 9 materials-13-02136-f009:**
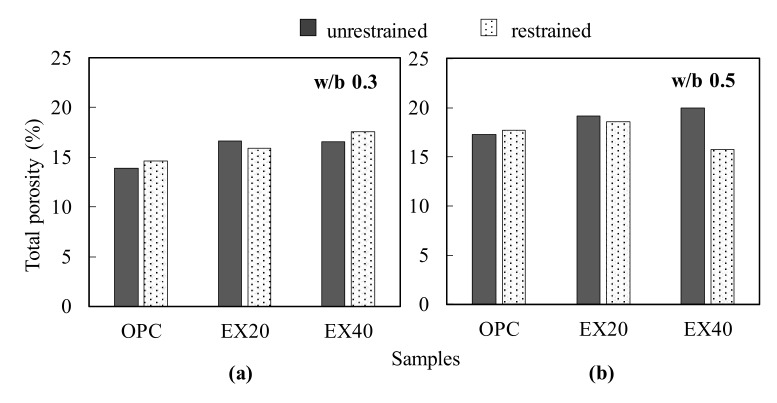
Relationship between the total porosity and EX contents under the unrestrained and triaxial restrained conditions: (**a**) w/b ratio of 0.3 and (**b**) w/b ratio of 0.5.

**Figure 10 materials-13-02136-f010:**
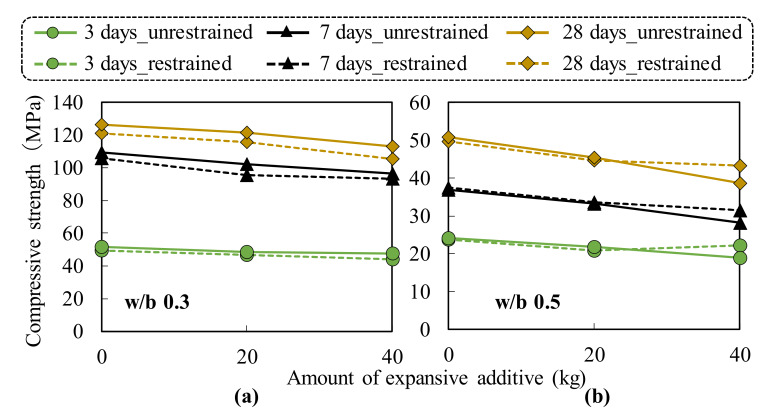
Relationship between the EX content and compressive strength of concrete under the unrestrained and restrained conditions: (**a**) w/b ratio of 0.3 and (**b**) w/b ratio of 0.5.

**Figure 11 materials-13-02136-f011:**
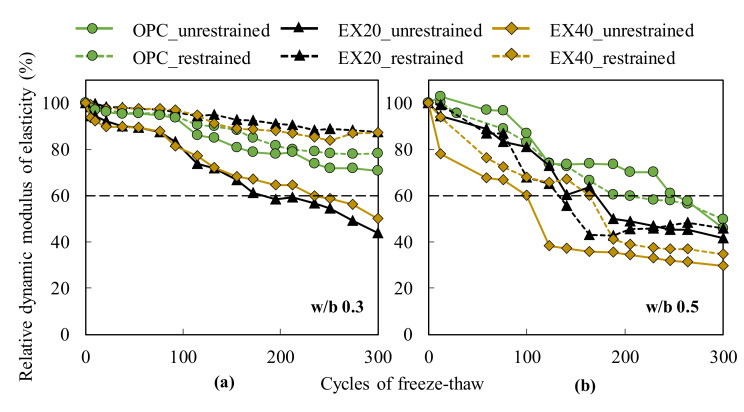
Comparison of the RDMs of cement and expansive concrete under the restrained and unrestrained conditions: (**a**) w/b ratio of 0.3 and (**b**) w/b ratio of 0.5.

**Figure 12 materials-13-02136-f012:**
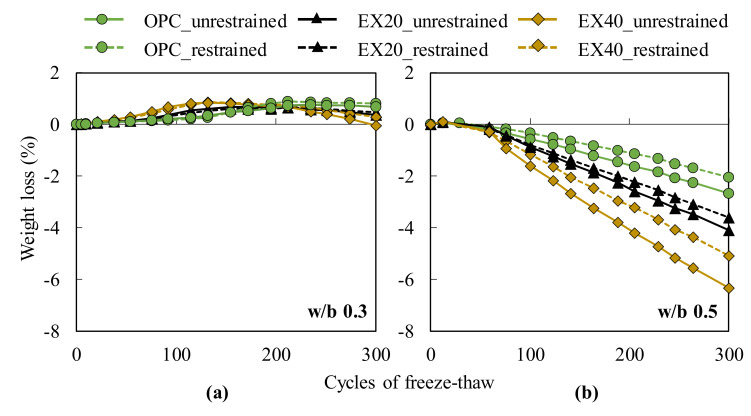
Mass change of all concrete mixtures during the freeze–thawing test: (**a**) w/b ratio of 0.3 and (**b**) w/b ratio of 0.5.

**Table 1 materials-13-02136-t001:** Material properties.

Materials	Symbol	Properties
Ordinary Portland cement	C	Density: 3.15 g/cm^3^; Specific surface area: 3490 cm^2^/g
Expansive additive	EX	Ettringite-gypsum type; Density: 3.05 g/cm^3^; Specific surface area: 3260 cm^2^/g
Coarse aggregate	G	Density: 2.68 g/cm^3^; Absorption: 2.17%
Fine aggregate	S	Density: 2.68 g/cm^3^; Absorption: 1.78%
Water	W	Tap water
Admixtures	Ads	w/b 0.5: AE water-reducing agent (Master Pozzolith No. 70) w/b 0.3: High-performance water-reducing agents (SP8SV) and AE agent (Master air 101)

**Table 2 materials-13-02136-t002:** Mix proportions.

Series	Symbol	w/b	Unit Weight, kg/m^3^					Ads
			W	C	EX	S	A	
Series 1	OPC	0.3	175	583	−	616	981	SP8SV (B × 1.2%) Master air 101 (B × 0.001%)
	EX20			563	20			
	EX40			543	40			
Series 2	OPC	0.5	185	370	−	855	959	No. 70 (250 mL/B = 100 kg)
	EX20			350	20			
	EX40			330	40			

Note: B: binder; W: water; C: cement; EX: expansive additive; S: sand; A: aggregate; Ads: admixtures.

**Table 3 materials-13-02136-t003:** Test results of fresh concrete.

Series	Sample	w/b	Slump (cm)	Flow (cm)	Air Content (%)	Temperature (°C)
Series 1	OPC	0.3	−	66	4.7	19.5
	EX20		−	65.5	5.5	19.5
	EX40		−	66	5.8	20
Series 2	OPC	0.5	20	−	4.8	15.0
	EX20		19.4	−	4.9	15.5
	EX40		19.5	−	4.9	16.2

**Table 4 materials-13-02136-t004:** Critical pore entry diameter (nm).

Sample	w/b 0.3		w/b 0.5	
	Unrestrained	Restrained	Unrestrained	Restrained
OPC	28.11	24.28	35.95	33.28
EX20	30.77	26.05	33.89	33.89
EX40	28.26	26.24	48.38	40.81

**Table 5 materials-13-02136-t005:** Carbonation depth of concrete (mm).

Sample	w/b 0.3		w/b 0.5	
	Unrestrained	Restrained	Unrestrained	Restrained
OPC	0	0	1.15	0.95
EX20	0	0	2.90	1.78
EX40	0	0	3.60	2.80

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
