# Peer review of "Influence of Restrained Condition on Mechanical Properties, Frost Resistance, and Carbonation Resistance of Expansive Concrete"

_materials, 2020, doi:10.3390/ma13092136_

Round 1
Reviewer 1 Report
Questions to author,
Q1: line 18, abstract should be clear and concise. As suitable are the line 72-78, I recommended to author overwrite the abstract
Q2: line 60- 61, no correct write index cm3
Q3: Table 1, in Table 1 is not unit of specific gravity; in Table 1 are determined the same the value of density aggregates (fine and coarse)? I recommend to state the granulometry of these materials
Q4: Table 2, under the names of used raw materials, I would recommend adding the indices (abbreviations) G, S, W, etc. Furthermore, I might extend Table 2 to include the tests that were performed on individual samples.
Q5: line 109, what do you author mean „according to the free standard“
Q6: line 130, add units for formulas, I also recommend making larger spaces between formulas, it is confusing, in the text are anywhere references of formulas
Q7: line 136, why the compressive strength on the rollers was measured instead of on the cubes
Q8: line 166, no correct link to the table
Q9: line 166, here the author compares the measured values ​​of “Flow” for (Series 1) and “Slump” for (Series 2) why not all values ​​are measured? How does temperature affect other parameters in this case, when it is so variable in a given measurement.
Q10: Figure 5, no correct marked pictures
Q11: line 232-234, The first hypothesis is relevant to the formation of the ettringite crystal when the expansive additive reacts with water, which will fill the microspore in the cement–expansive paste matrix. This phenomenon leads to a denser pore structure. This hypothesis was also reported by Van et al. [22] and Nguyen et al. [2]. Tato hypotéza byla zmínÄ›na i v závÄ›ru. This hypothesis was mentioned in the conclusion. Why the author did not confirm the presence of etringite.
Q12: line 255, the text is very descriptive and there are no more precise values ​​of compressive strength, let the author state for the experiment exactly measured values ​​and not only approximate
Q13: line 254, In conclusion, observing the effect of the restrained condition on the total porosity was easier with a lower stiffness in the expansive concrete. You can author expalin the term „porosity was easier“
Q14: line 260, I recommend the use unit MPa
Q15: Figure 10, edit the legend, as on the other Figures in the experiment
Q16: line 314, Please add the pH value, which in this case is also important, as the color of the concrete changed in the experiment
Q17: line 336,. The conclusion as again very general and descriptive.
Reviewer 2 Report
The paper Influence of restrained condition on mechanical properties, frost resistance, and carbonation resistance of expansive concrete by Duc Van Nguyen, Emika Kuroiwa, Jihoon Kim, Hyeonggil Choi, Yukio Hama is well suited for journal Materials. The authors of this article analyzed the results of the present studies on the effect of the restrained condition on the mechanical properties, frost resistance, and carbonation resistance of expansive concrete with different water-binder ratios.
The paper is interesting, presents research and comprehensive elaboration of results. The paper contains parts in good order: introduction, experimental program, results and discussion, conclusions. It should be emphasized that the text and drawings are very well prepared.
The length of the abstract is good, enough to put a lot of information, summarize the article, but not too long for the reader who wants to get information about the content of the article and decide whether to read the whole. A general but basic conclusion is given "The results demonstrate that the restrained condition can enhance the mechanical properties and the durability of the expansive concrete." The interested reader, wanting to get more facts must download the entire article, will not stop at the abstract.
Introduction length, about one page, seems good. It starts with showing the "drying shrinkage" problem, the rest shows a solution to this problem in literature. Literature list rather short for current standards, 3 articles from the last 2 years. The authors in the context of previous studies described in the literature indicate the role of their article "Note that most of these works only focused on the mechanical behavior of expansive concrete under uniaxial restrained conditions. Meanwhile, an experimental study on the effect of both uniaxial and triaxial restrained conditions on the mechanical properties and durability of expansive concrete has not yet been established. "
The experimental program is described correctly. Materials and mix proportions are described in detail. Test methods have been described, but also referred to by standards, which will allow reproduce the experiments and get the same outcomes. The drawings showing samples and experiments were prepared very well, with sufficient detail and legibility (only what is necessary, not too much, not too little). The colors in the drawings are used to highlight information, they are not too many.
Discussion of the results was placed with a description of the results. In the case of a description of many related experiments, it is convenient for the reader, he obtains information about the partial experiment, results and their explanation presented by the authors in the context of the results of other partial experiments.
Conclusions are bulleted, which makes it easier for readers to extract information in line with the authors' expectations, as well as to group knowledge.
The article was written enough well in English, is understandable for a reviewer, a person who does not speak English as a mother tongue.
For the reasons stated, I support publication of the paper in the journal.
Reviewer 3 Report
The Abstract needs to be more factual and present the most important contribution of the study to the readers.
The introduction should provide wider literature review for the concrete protection: e.g: https://doi.org/10.1016/j.conbuildmat.2019.117057 ; https://doi.org/10.1016/j.rineng.2020.100110
The objectives of the paper should also reflect the bigger aim of the study. How are the achievement of objectives contributing to the main goal? what is the current problem? and how the designed study will solve or contribute to solving a real problem?
The discussions and reasoning for the observed phenomenons are well presented.
The conclusions, can also be a little more concise.
Round 2
Reviewer 1 Report
Dear author, I have only one little reminder,
at Q15: please delete the frame by legend". The colours is OK.
Thank you for your answers.